

# 1 Changes of regional meteorology induced by anthropogenic heat

# 2 and their impacts on air quality in South China

Min Xie[1,2*], Kuanguang Zhu[1,3], Tijian Wang[1,4*], Wen Feng[2], Minggao Li[3], Mengmeng Li[1], Yong
Han[1], Shu Li[1], Bingliang Zhuang[1], Lei Shu[1], Da Gao[1], Jingbiao Liao[1]
[1] School of Atmospheric Sciences, Nanjing University, Nanjing, China;
[2] Key Laboratory of South China Sea Meteorological Disaster Prevention and Mitigation of
Hainan Province, Haikou, China
[3] Hubei Academy of Environmental Science, Wuhan, China
[4] CMA-NJU Joint Laboratory for Climate Prediction Studies, Institute for Climate and Global
Change Research, School of Atmospheric Sciences, Nanjing University, Nanjing, China
**Abstract:** Anthropogenic heat (AH) emissions from human activities can change the urban
circulation and thereby affect the air pollution in and around cities. Based on statistic data, the
spatial distribution of AH flux in South China is estimated. With the aid of the WRF/Chem model
in which the AH parameterization is developed to incorporate the gridded AH emissions with
temporal variation, the simulations for January and July in 2014 are performed over South China.
By analyzing the differences between the simulations with and without adding AH, the impact of
AH on regional meteorology and air quality are quantified. The results show that the regional
annual mean AH fluxes over South China are only 0.87W/m$^2$, but the values for the urban areas of
the Pearl River Delta (PRD) region can be close to 60 W/m$^2$. These AH emissions can
significantly change the urban heat island and urban-breeze circulations in the big cities. In the
PRD city cluster, 2-m air temperature rises up by 1.1℃ in January and over 0.5℃ in July, the
boundary layer height increases by 120m in January and 90m in July, 10-m wind speed is
intensified over 0.35 m/s in January and 0.3 m/s in July, and the accumulative precipitation is
enhanced by 20-40% in July. These changes of meteorological conditions can significantly impact
the spatial and vertical distributions of air pollutants. Due to the increases of PBLH, surface wind
speed and upward vertical movement, the concentrations of primary air pollutants decrease near
surface and increase at the upper levels. But the vertical changes of O$_3$ concentrations show the
different patterns in different seasons. The surface O$_3$ concentrations in big cities increase with
maximum values over 2.5ppb in January, while O$_3$ is reduced at the lower layers and increases at
the upper layers above some megacities in July. This phenomenon should be attributed to the facts
that the chemical effects can play a significant role in O$_3$ changes over South China in winter,
while the vertical movement can be the dominant effect in some big cities in summer. Adding the
gridded AH emissions can better describe the heterogeneous impacts of AH on regional
meteorology and air quality, suggesting that more studies on AH should be carried out in the
climate and air quality assessments.
**Key words:** Anthropogenic heat; PRD; WRF/Chem; PM$_{10}$; O$_3$






---------------------------------------------------------------------------------------------------

∗Corresponding author, +86-25-89685302
E-mail address: minxie@nju.edu.cn, tjwang@nju.edu.cn

Urbanization and its impacts on regional meteorology and air quality have been widely
acknowledged, observed, and investigated (Rizwan et al., 2008; Mirzaei and Haghighat, 2010).
Previous studies have illustrated that urbanization can affect atmospheric environment in many
ways, which are mainly associated with the increase of air pollutant emissions from the
intensification of energy consumptions (Akbari et al., 2001; Civerolo et al., 2007; Jiang et al, 2008;
Stone, 2008; Chen et al., 2014b), the change of land covers from natural surfaces to artificial ones
(Civerolo et al., 2007; Lo et al., 2007; Wang et al., 2007; 2009b; Jiang et al., 2008; Zhang et al.,
2009; Lu et al., 2010; Wu et al., 2011; Chen et al., 2014b; Liao et al., 2015; Zhu et al., 2015; Li et
al., 2016), and the release of anthropogenic heat from human activities in cities (Ryu et al., 2013;
Yu et al., 2014; Xie et al., 2016). Anthropogenic heat (AH) can increase turbulent fluxes in
sensible and latent heat (Oke, 1988), implying that it can modulates local and regional
meteorological processes (Ichinose et al., 1999; Block et al., 2004; Fan and Sailor, 2005; Ferguson
and Woodbury, 2007; Chen et al., 2009; Zhu et al., 2010; Feng et al., 2012; 2014; Menberg et al.,
2013; Ryu et al., 2013; Wu and Yang, 2013; Bohnenstengel et al., 2014; Chen et al., 2014a; Meng
et al., 2011; Yu et al., 2014; Xie et al., 2016) and thereby exert an important influence on the
formation and the distribution of ozone (Ryu et al., 2013; Yu et al., 2014; Xie et al., 2016) as well
as aerosols (Yu et al., 2014; Xie et al., 2016).
Previous studies on AH basically focused on the amount of heat fluxes or their effects on
meteorology. It was reported that the typical values of AH fluxes in urban areas range from 20 to
100 W/m$^2$ (Crutzen, 2004; Sailor and Lu, 2004; Fan and Sailor, 2005; Pigeon et al., 2007; Lee et
al., 2009; Iamarino et al., 2012; Lu et al., 2016; Xie et al., 2016). Sometimes, the fluxes might
exceed the value of 100 W/m$^2$ (Iamarino et al., 2012; Quah and Roth, 2012; Lu et al., 2016; Xie et
al., 2016), with the extreme value of 1590 W/m$^2$ in the densest part of Tokyo at the peak of
air-conditioning demand (Ichinose et al., 1999). In regard to their effects, the researchers found
that AH fluxes can cause urban air temperatures to increase by several degrees (Fan and Sailor,
2005; Ferguson and Woodbury, 2007; Chen et al., 2009; Zhu et al., 2010; Feng et al., 2012; 2014;
Menberg et al., 2013; Wu and Yang, 2013; Bohnenstengel et al., 2014; Chen et al., 2014a; Yu et al.,
2014; Xie et al., 2016), induce the atmosphere more turbulent and unstable, change the urban heat
island circulation, strengthen the air vertical movement (Ichinose et al., 1999; Block et al., 2004;
Fan and Sailor, 2005; Chen et al., 2009; Feng et al., 2012; 2014; Bohnenstengel et al., 2014; Yu et
al., 2014; Xie et al., 2016), enhance the convergence of water vapor in cities, and change the
regional precipitation patterns (Feng et al., 2012; 2014; Xie et al., 2016). In spite that meteorology
conditions and air quality are inextricably linked, however, few investigations have paid attention



to how the air quality is altered by the changes of regional meteorology induced by anthropogenic
heat. The results from the limited studies have showed that this impact is significant in and around
large urban areas and should be considered in the air pollution predictions (Ryu et al., 2013; Yu et
al., 2014; Xie et al., 2016).
Over the past decades, many areas in South China have been suffering the air quality
deterioration (Wang et al., 2007; 2009b; Chan and Yao, 2008; Liu et al., 2013b), with high ozone
($O_3$) or poor visibility frequently occurring in urban areas (Wang et al., 2007; Fang et al., 2009)
and the background air pollutant concentrations steadily increasing (Wang et al., 2009a; Liu et al.,
2013b). These air pollutions may be related with the rapid urban expansion. As the most urbanized
and industrialized part of South China, the Pearl River Delta (PRD) region has become the largest
metropolitan area in the world within a very short time (Word Bank Group, 2015). Thus, many
previous studies have tried to figure out the effects of urbanization on urban climate and air quality
in this region (Lo et al., 2007; Wang et al., 2007; 2009b; Lu et al., 2010; Meng et al., 2011; Wu et
al., 2011; Zhang et al., 2011; Feng et al., 2012; 2014; Chen et al., 2014b; Li et al., 2014; 2016).
Among these studies, most researchers merely investigated how the expansion of urban land-use
influences the meteorology processes (Lo et al., 2007; Wang et al., 2007; 2009b; Lu et al., 2010;
Meng et al., 2011; Wu et al., 2011; Feng et al., 2012; Chen et al. 2014b; Li et al., 2016). Some also
linked these changes of meteorological factors with the regional air quality, and quantified the
impacts of land-use changing on air pollution (Wang et al., 2007; 2009b; Feng et al., 2012; Chen
et al., 2014b; Li et al., 2014; 2016). Only a few researchers took AH into account (Meng et al.,
2011; Feng et al., 2012; 2014). But they just clarified the impact of AH on meteorological
conditions by merely adopting the fixed AH value in the urban parameterization scheme of
meteorological models (Meng et al., 2011; Feng et al., 2012). Consequently, we still need to
further understand how the excessive anthropogenic heat from urban expansion impacts on the
severe air quality problems in this world famous region.
To fill the abovementioned knowledge gap, we present our new findings on the impact
mechanism of anthropogenic heat on urban climate and regional air quality over South China in
this paper, including (1) the spatial and temporal characteristics of AH emissions in South China,
(2) how to implement the inhomogeneous AH data into the air quality model WRF/Chem, (3) the
impacts of AH fluxes on meteorological fields, and (4) the impacts of meteorology changes on the
air quality in different cities over South China. Detailed descriptions about the estimating method
for anthropogenic heat emissions, the adopted WRF/Chem model with special configurations, and
the observation data for model validation are presented in Sect. 2. Main results, including the
inhomogeneous distribution of AH, the model evaluation, and the three-dimensional changes of
meteorological fields and air pollutant concentrations are presented in Sect. 3. The summary is
given in Sect. 4.

**2. Methodology and data**





### 2.1 Method for estimating anthropogenic heat fluxes

The top-down energy inventory method, which predicts AH emissions based on the statistics data of energy consumption, is the most common approach and widely used all over the world (Sailor and Lu, 2004; Flanner, 2009; Hamilton et al., 2009; Lee et al., 2009; Allen et al., 2011; Iamarino et al., 2012; Quah and Roth, 2012; Chen et al., 2014a) as well as in China (Chen et al., 2012; Xie et al., 2015; 2016; Lu et al., 2016). On basis of the previous studies, AH fluxes over the area between (101°E, 16°N) and (119°E, 26°N) in 1990, 1995, 2000, 2005, 2010 and 2014 are calculated in this study by the following equation:

$$Q_F = Q_{F,I} + Q_{F,B} + Q_{F,V} + Q_{F,HM} \tag{1}$$

where, $Q_F$ is the total anthropogenic heat flux (W/m$^2$); $Q_{F,I}$, $Q_{F,B}$, $Q_{F,V}$, and $Q_{F,HM}$ represent the heat emitted from the industry sector, buildings, vehicles and human metabolism (W/m$^2$), respectively. To accurate estimate the spatial heterogeneity of AH fluxes, the estimated area is gridded as 456 rows and 264 columns with the grid spacing of 2.5 arcmin. The heat flux generated by human metabolism at each grid is estimated as:

$$Q_{F,HM} = P \cdot (M_d \cdot h_d + M_n \cdot h_n) / h \tag{2}$$

where, $P$ is the population number at a grid. $h_d$, $h_n$ and $h$ are the hours of daytime, nighttime and a whole day. In this study, they are set to be 16, 8 and 24, respectively. $M_d$ and $M_n$ are the average human metabolic rate (W/person) during the daytime and at night. Referring to the previous studies (Sailor and Lu, 2004; Chen et al., 2012; Quah and Roth, 2012; Xie et al., 2015; 2016; Lu et al., 2016), we determined that the metabolic rate of a typical man is 175 W for the active daytime ($M_d$) and 75 W for the sleep period ($M_n$).

Based on the work of Flanner (2009), Lu et al. (2016) and Xie et al. (2016), it is reasonably assumed that all non-renewable primary energy consumption used for human activities is thermally dissipated as AH. So, $Q_{F,I}$, $Q_{F,B}$, and $Q_{F,V}$ at each grid can be estimated by using the data of non-renewable energy consumption (coal, petroleum, natural gas, and electricity etc.) from different categories. The amount of AH fluxes for one category can be estimated by the following equation:

$$Q_x = \eta \cdot \varepsilon \cdot C / (t \cdot A) \tag{3}$$

where, $Q_x$ represents $Q_{F,I}$, $Q_{F,B}$ or $Q_{F,V}$. $C$ is the primary energy consumption from a category at a grid (metric ton standard coal). $\varepsilon$ is the calorific value of standard coal equivalent, with the recommended value of $29.271 \times 10^3$ kJ/kg (Chen et al., 2012; Lu et al., 2016; Xie et al., 2015; 2016). $\eta$ is the efficiency of heat release, with the typical value of 60% for electricity or heat-supply sector and 100% for other sectors (Lu et al., 2016; Xie et al., 2016). $t$ is the time duration of used data, which is set to be 31536000 s (seconds in a year) in this study. $A$ represents the area of a grid (km$^2$). To quantify the value of $C$ for each grid, we first of all obtain the energy consumption data from 1990 to 2014 in China Energy Statistical Yearbooks. Then we double check and modify the data in typical cities on basis of the Yearbooks in Guangdong, Guangxi, Hainan province and Hong Kong. In the end, the total numbers are apportioned according to the





value of gross domestic product (GDP) or population density at each grid. GDP is used for
industry and vehicle, while population is chosen for building. The population density with the
resolution of 2.5 arcmin in 1990, 1995, 2000, 2005 and 2010 can be downloaded from Columbia
University's Socioeconomic Data and Applications Center. The gridded GDP data are developed
and applied based on the work of Liu et al. (2013a). The spatial distributions of GDP and
population in 2014 are unobtainable, and thereby the data in 2010 are used as the surrogates.
**2.2 WRF/Chem and its configuration**
The WRF/Chem version 3.5 is applied to investigate the impacts of AH fluxes on regional
meteorology and air quality over South China. WRF/Chem is a new generation of air quality
modeling system, in which the feedbacks between meteorology and air pollutants are included by
fully coupling the meteorological model (WRF) with the chemical modules (Chem). WRF/Chem
has been widely used in simulating air quality in China and proved to be a reliable modeling tool
from city-scale to meso-scale (Wang et al., 2009b; Liu et al., 2013; Yu et al., 2014; Liao et al.,
2015; Xie et al., 2016).
Three simulations are conducted in this study. One does not take the contribution of AH into
account while the other two incorporate WRF/Chem with the fixed or the inhomogeneous AH
fluxes (The details are presented in Sect. 2.3). Except for the setting of AH parameterization, other
configurations (such as the physical schemes, the chemical schemes and the emission inventories
etc.) for all simulations are the same. Thus, the difference between the modeling results can
illustrate the effects of AH. As shown in Fig. 1, two nested domains are used. The outermost
domain (Domain 1, D01) has the horizontal grids of $121 \times 95$, with the grid resolution of 27km $\times$
27km. The second domain (Domain 2, D02) covers Guangdong, Guangxi, and Hainan provinces,
with the center point at (110.4°E, 20.9°N), the horizontal grids of $192 \times 105$, and the grid spacing
of 9km. For all domains, from the ground level to the top pressure of 100hPa, there are 31 vertical
sigma layers with about 10 in the planetary boundary layer (PBL). January and July in 2014 are
chosen for simulations and analysis. January and July are used to represent the hot and the cold
weather condition, respectively.

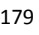

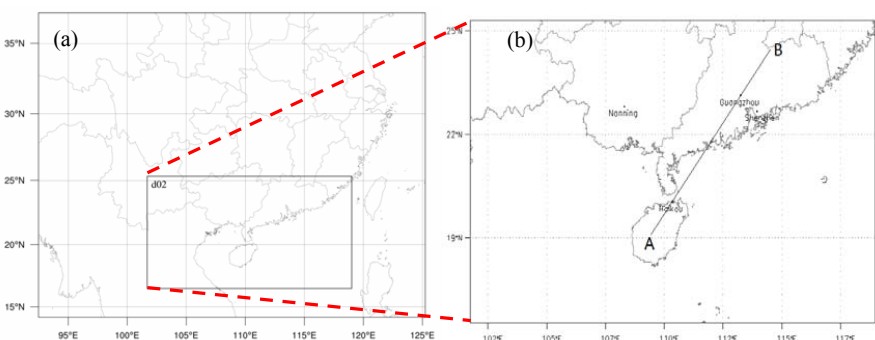


**Fig. 1. WRF/Chem domain configuration: (a) tow domains for simulations and (b) enlarged view of domain**



**2 with the cities in South China where the observation sites are located. Line AB in (b) denotes the location**
**of the vertical cross section used in Fig. 4, Fig. 6, Fig. 8, Fig. 9, and Fig. 10.**

The detailed options for the physical and chemical parameterization schemes used in this
study are shown in Table 1. Additionally, a Single Layer Urban Canopy Model (SLUCM) coupled
in Noah Land Surface Model (Noah/LSM) is adopted for better modeling the urban effects.
Following the work of Liu et al. (2013), the default values for urban canopy parameters in
SLUCM are substituted by the typical values in South China. The recently updated Moderate
Resolution Imaging Spectroradiometer (MODIS) land-use data (20 categories) with the resolution
of 30-sec are used to replace the default USGS (U.S. Geological Survey) land-use data in
WRF/Chem, because the USGS data are too outdated to illustrate the intensive urbanization over
South China. For chemistry, the RADM2 gas-phase chemistry scheme and the MADE/SORGAM
aerosol scheme are adopted. RADM2 (Regional Acid Deposition Model version 2) contains 63
prognostic species and 136 reactions (Balzarini et al., 2015). MADE/SORGAM is the classical
aerosol module used in WRF/Chem (Grell et al., 2005), where the Aerosol Dynamics Model for
Europe (MADE) (Ackermann et al., 1998) contains the Secondary Organic Aerosol Model
(SORGAM) (Schell et al., 2001). The anthropogenic emissions are mainly from the 2012-year
Multi-resolution Emission Inventory for China (MEIC) with $0.25° \times 0.25°$ resolution. This MEIC
inventory based on RADM2 mechanism is re-projected for the grids of China in both domains.
For the grids outside of China, the inventory developed by Zhang et al. (2009) is used. The
biomass burning emissions are acquired from the work of Li et al. (2016). The biogenic emissions
are calculated online by using MEGAN2.04 (Guenther et al., 2006). The NCEP global reanalysis
data with the spatial resolution of $1° \times 1°$ and 27 vertical levels are selected to provide the initial
meteorological fields and boundary conditions. The initial chemical state and boundary conditions
are obtained from the modeling results from the global chemistry transport model MOZART-4.

**Table 1. The grid settings, physics and chemistry options for all simulations**

| Items | Contents |
|---|---|
| Dimensions (x,y) | (121,95), (192,105) |
| Grid size (km) | 27, 9 |
| Time step (s) | 150 |
| Microphysics | Purdue Lin microphysics scheme (Lin et al., 1983) |
| Long-wave radiation | RRTM scheme (Mlawer et al., 1997) |
| Short-wave radiation | Goddard scheme (Kim and Wang, 2011) |
| Cumulus parameterization | Grell 3D (Grell and Devenyi, 2002) |
| Surface layer | Eta similarity (Janjic, 1994) |
| Land surface | Noah land surface model (Chen and Dudhia, 2001) |
| Planetary boundary layer | Mellor-Yamada-Janjic scheme (Janjic, 1994) |
| Gas-phase chemistry | RADM2 (Stockwell et al., 1990) |
| Photolysis scheme | Madronich photolysis (Madronich, 1987) |
| Aerosol module | MADE (Ackermann et al., 1998) / SORGAM (Schell et al., 2001) |


**2.3 The configurations for AH parameterization**
As shown in Table 2, three cases of numerical experiments are performed to evaluate the
effects of AH. Non_AH is the base case, which does not consider the effects of AH. In Fix_AH,





the default option for AH in SLUCM of WRF/Chem is adopted. For Grd_AH, we modify the AH
parameterization, and the gridded AH flux data estimated in Sect. 2.1 are used to simulation the
spatial heterogeneous effects of AH on meteorology and air quality. The difference between the
modeling results of Fix_AH and Grd_AH can illustrate the model improvement caused by
considering the spatial heterogeneity of AH. Comparing the results from Non_AH and Grd_AH,
we can finally demonstrate the exact impacts of anthropogenic heat.

**Table 2. Three simulations conducted in this study**

| Cases | Description |
|---|---|
| Non_AH | excluding anthropogenic heat emissions in SLUCM |
| Fix_AH | including anthropogenic heat emissions in SLUCM, but using the default AH option with fixed value 50 W/m$^2$ for all urban grids |
| Grd_AH | including anthropogenic heat emissions in SLUCM, and using the inhomogeneous AH emissions in 2014 estimated in Sect. 2.1 |


In SLUCM of WRF/Chem, the AH for one grid is determined by the fixed AH value, the
fixed temporal diurnal pattern, and the urban fraction value (Chen et al., 2011; Yu et al., 2014; Xie
et al., 2016). This default parameterization for AH can be described by the following algorithm:
$\qquad SH = F_V \cdot SH_V + F_U \cdot (SH_U + AH_{fixed})$ $\qquad\qquad\qquad$ (4)
where $SH$ is the total sensible heat flux in a grid. $F_V$ and $SH_V$ are the fractional coverage and the
sensible heat flux of vegetations, respectively. $F_U$ and $SH_U$ are those of urban surfaces. $AH_{fixed}$
represents the fixed AH value for all urban areas (Chen et al., 2011). With respect to Grd_AH, we
modify Eq. 4 by incorporating the inhomogeneous AH data ($Q_F$) as follow:
$\qquad SH = F_V \cdot SH_V + F_U \cdot (SH_U + Q_F)$ $\qquad\qquad\qquad\qquad$ (5)
The gridded AH fluxes in 2014 from Sect. 2.1 (with the grid spacing of about 4km) are
re-projected to domain 2 (9km) by the coordinates of each grid. To account for temporal variability,
the diurnal variation pattern recommended for PRD by Zheng et al. (2009) and Lu et al. (2016) is
adopted. It was reported that there is no significant seasonal difference in heating over South
China (Lu et al., 2016). Thus, the monthly variation of AH is not considered in this study.
**2.4 Method for model evaluation**
The observation data of meteorology factors and air pollutants in Guangzhou, Shenzhen,
Nanning and Haikou are used to validate the WRF/Chem simulations in this study. The hourly
observation records of 2-m temperature, 10-m wind speed and 2-m relative humidity in January
and July of 2014 can be obtained from the National Meteorological center of China
Meteorological Administration. The relevant time series of $PM_{10}$ and $O_3$ concentrations can be
acquired from China National Environmental Monitoring Center. The assurance/quality control
(QA/QC) procedures for these data strictly follow the national standards. As described by Liao et
al. (2015) and Xie et al. (2016), the mean bias (MB), root mean square error (RMSE) and
correlation coefficient (COR) between observation records and modeling results are used to
evaluate the model performance.





## 3. Results and discussions

### 3.1 Spatial distribution of AH fluxes in South China

Fig. 2 shows the spatial distribution of AH in 1990, 1995, 2000, 2005, 2010 and 2014 over South China. Obviously, big cities especially the cities in PRD have the largest values from the 1990s till now. In 1990, except for those in Guangdong and Hong Kong, the AH fluxes in most areas of South China are less than 2 $W/m^2$. From 1995 to 2000, the AH fluxes in most parts of PRD (except for those in Hong Kong) are less than 5 $W/m^2$, and those in other areas of South China are generally lower than 2.5 $W/m^2$. After 2005, however, the AH fluxes exceed 10 $W/m^2$ in many cities of South China, with the high values over 50 $W/m^2$ in and around Hong Kong. For the annual mean AH flux over the whole administrative district of different province, the value in Guangdong continuously increases from 0.30 $W/m^2$ for 1990 to 1.68 $W/m^2$ for 2014, while the heat release in Guangxi and Hainan keeps in a low level (< 0.5 $W/m^2$) but with an obvious increasing. The annual mean AH values in the downtown areas are much higher than the regional ones. For instance, the PRD city cluster always has the highest anthropogenic heat emissions in South China. As shown in Table 3, the annual mean value in the built-up areas aggrandizes from 5.1 $W/m^2$ in 1990 to 58 $W/m^2$ in 2014. These results are similar to those reported by Chen et al. (2012a; 2014) and Xie et al. (2015), and the temporal variation pattern also fits in well with the economic boom over South China in the past decades.

In 2014, as illustrated in Fig. 4f, most important cities in South China have the AH fluxes more than 5 $W/m^2$. High fluxes generally occur in Guangdong province, especially in the PRD region and the Chao-Shan area, with the typical values over 10 $W/m^2$. In the build-up area of Guangzhou, the AH fluxes are close to 60 $W/m^2$, which are similar to those in Seoul of Korea (Lee et al., 2009), Toulouse of France (Pigeon et al., 2007), and some US cities (Sailor and Lu, 2004; Fan and Sailor, 2005). The regional highest value occurs in Hong Kong, with the value exceeding 100 $W/m^2$. This value is comparable to those in the most crowded megacities, such as Shanghai (Xie et al., 2016), Tokyo (Ichinose et al., 1999), London (Hamilton et al. 2009; Iamarino et al. 2012), and Singapore (Quah and Roth, 2012). In Nanning and Haikou, the annual mean AH fluxes over the whole administrative district are close to 10 $W/m^2$. Our spatial distribution of AH based on the population reflects the economic activities in South China, suggesting that our method is effective and the results are reasonable. These results can be supported by other previous investigations (Flanner, 2009; Chen et al., 2012a; 2014; Xie et al., 2015; Lu et al., 2016). So, our AH data can be used in models to investigate their impacts on urban climate and air quality.


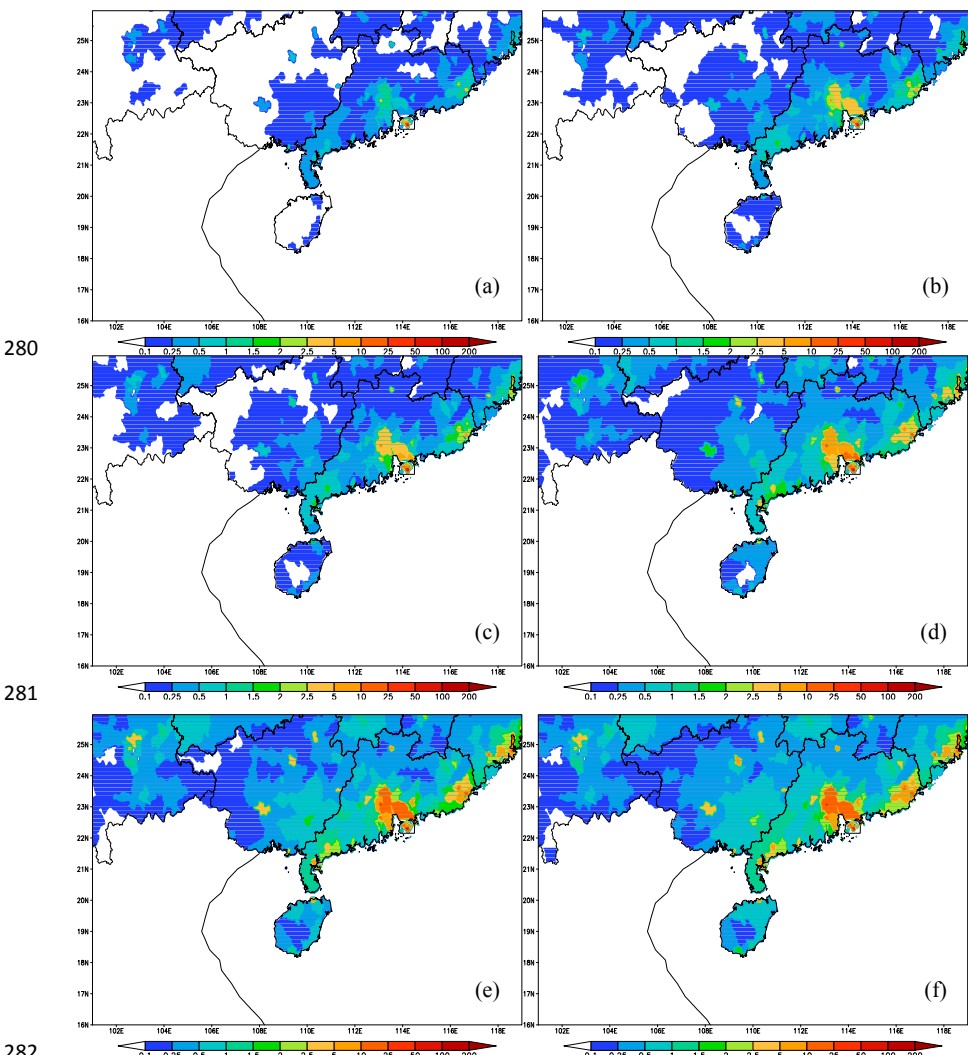

**Fig. 2. Annual-mean anthropogenic heat fluxes between (101°E, 16°N) and (119°E, 26°N) with the resolution of 2.5 arcmin in 1990 (a), 1995 (b), 2000 (c), 2005 (d), 2010 (e) and 2014 (f), respectively.**

**Table 3 Annual average anthropogenic heat flux in different administrative district over South China (W/m$^2$)**

| Province | | This study | | | | | |
|---|---|---|---|---|---|---|---|
| | | 1990 | 1995 | 2000 | 2005 | 2010 | 2014 |
| Guangdong | Regional [a] | 0.30 | 0.48 | 0.61 | 1.05 | 1.53 | 1.68 |
| | Urban area in PRD | 5.11 | 11.13 | 14.51 | 30.82 | 49.41 | 58.03 |
| Guangxi | Regional [a] | 0.11 | 0.16 | 0.17 | 0.26 | 0.38 | 0.44 |
| Hainan | Regional [a] | 0.04 | 0.09 | 0.14 | 0.23 | 0.37 | 0.49 |

[a] Regional represents the average value over the whole area of a province

**3.2 Simulation performance**




To evaluate the model performance and clarify the better AH parameterization, the modeling
results from Fix_AH and Grd_AH are compared with the observation data in two typical months
(January and July). Table 4 presents the performance statistics, including the values of monthly
mean (Mean), mean bias (MB), root mean squared error (RMSE) and correlative coefficient
(COR), which are all quantified for 2-m temperature ($T_2$), 2-m relative humidity ($RH_2$), 10-m wind
speed ($WS_{10}$), ozone ($O_3$), and particles ($PM_{10}$) in Guangzhou (GZ), Shenzhen (SZ), Nanning
(NN), and Haikou (HK).
As shown in Table 4, the correlation coefficients (COR) between observations and
simulations at four sites are generally about 0.80 for $T_2$, over 0.75 for $RH_2$, and close to 0.70 for
$WS_{10}$ in both January and July (statistically significant at the 95 % confident level). So adding AH
in WRF/Chem (Fix_AH and Grd_AH) can well describe the urban meteorological conditions in
the typical cities over South China. Compared with the observation records of $T_2$, except for
Shenzhen in January, both Fix_AH and Grd_AH tend to slightly simulate higher 2-m air
temperature at four sites in both months, which can be attributed to the uncertainty of urban
canopy and surface parameters (Liao et al., 2015; Xie et al., 2016). These overestimates are
acceptable because the MB values are smaller than 1.8 ℃ in January and smaller than 0.8 ℃ in
July. Moreover, when the gridded AH fluxes are taken into account (Grd_AH), the modeling
results of air temperature can be improved, with the mean bias (MB) decreasing by 0.1 - 0.3 ℃
and the correlation coefficient (COR) increasing by 0.02 - 0.05 (from Fix_AH to Grd_AH). With
regards to $RH_2$, the modeling values from two simulations (Fix_AH and Grd_AH) are close to the
observations. The best simulation occurs in Haikou, and the results at the other three sites are
reasonable as well, only with the bias within ±10%. These 2-m relative humidity predictions can
be improved from Fix_AH to Grd_AH. When we consider the heterogeneity of AH fluxes in
Grd_AH, the values of MB and RMSE are closer to 0 and those of COR are closer to 1. For $WS_{10}$,
because the modeling near-surface wind speed is generally influenced by local underlying surface
characteristics more than other meteorological parameters (Liao et al., 2015; Xie et al., 2016), both
Fix_AH and Grd_AH slightly overvalue the 10-m wind speed at four sites. In case Fix_AH, the
MB for $WS_{10}$ is generally around 1m/s in both months, and the RMSE is less than 2.6 m/s in
January and around 2m/s in July. However, the predictions are obviously improved in case
Grd_AH. The MB decreases to 0.4-0.9 m/s in January and 0.4-0.7 m/s in July, and the values of
COR also increase from 0.68 (Fix_AH) to 0.74 (Grd_AH) in July. These improvements from
Fix_AH to Grd_AH for $T_2$, $RH_2$ and $WS_{10}$ predictions suggest that the default value of
WRF/Chem for all urban grids overestimates the AH fluxes in these cities, and our gridded AH
data as well as the new parameterization scheme can exactly catch the heterogeneity of the heat
released from the metropolitans of South China.



**Table 4 Summary of statistics for comparison between simulated and observed hourly averaged meteorological and chemical data in four cities of South China**

| Vars[a] | Site[b] | Fix_AH | | | | | | | | | | Grd_AH | | | | | | | | | |
|---|---|---|---|---|---|---|---|---|---|---|---|---|---|---|---|---|---|---|---|---|---|
| | | January | | | | | July | | | | | January | | | | | July | | | | |
| | | SIM[d] | OBS[e] | MB | RMSE | COR[f] | SIM[d] | OBS[e] | MB | RMSE | COR[f] | SIM[d] | OBS[e] | MB | RMSE | COR[f] | SIM[d] | OBS[e] | MB | RMSE | COR[f] |
| $T_2$ (°C) | GZ | 14.0 | 12.2 | 1.8 | 3.1 | 0.75 | 29.0 | 28.4 | 0.6 | 4.0 | 0.72 | 13.8 | 12.2 | 1.6 | 2.9 | 0.78 | 28.8 | 28.4 | 0.4 | 2.1 | 0.76 |
| | HK | 18.9 | 17.3 | 1.6 | 2.0 | 0.79 | 29.0 | 28.4 | 0.6 | 1.7 | 0.79 | 18.5 | 17.3 | 1.3 | 1.8 | 0.81 | 28.9 | 28.4 | 0.5 | 1.6 | 0.83 |
| | NN | 13.9 | 12.2 | 1.7 | 2.9 | 0.84 | 28.0 | 27.7 | 0.3 | 2.5 | 0.77 | 13.7 | 12.2 | 1.4 | 2.7 | 0.86 | 27.9 | 27.7 | 0.2 | 2.0 | 0.81 |
| | SZ | 14.6 | 14.7 | -0.1 | 1.8 | 0.84 | 29.9 | 29.1 | 0.8 | 2.0 | 0.76 | 14.4 | 14.7 | -0.3 | 1.9 | 0.86 | 29.6 | 29.1 | 0.5 | 1.9 | 0.81 |
| $RH_2$ (%) | GZ | 64.2 | 73.5 | -9.3 | 18.5 | 0.74 | 68.4 | 78.8 | -10.4 | 17.9 | 0.73 | 66.8 | 73.5 | -6.7 | 16.8 | 0.75 | 71.3 | 78.8 | -7.5 | 16.8 | 0.76 |
| | HK | 75.6 | 78.2 | -2.5 | 8.5 | 0.77 | 80.6 | 81.0 | -0.3 | 7.8 | 0.80 | 77.0 | 78.2 | -1.1 | 8.2 | 0.84 | 81.4 | 81.0 | 0.4 | 7.7 | 0.86 |
| | NN | 69.3 | 77.9 | -8.6 | 18.2 | 0.74 | 87.7 | 83.5 | 4.2 | 8.8 | 0.79 | 72.3 | 77.9 | -5.6 | 17.7 | 0.75 | 86.5 | 83.5 | 3.0 | 8.9 | 0.81 |
| | SZ | 65.9 | 63.3 | 2.6 | 11.7 | 0.75 | 74.2 | 78.0 | -3.8 | 11.1 | 0.75 | 66.5 | 63.3 | 3.2 | 12.3 | 0.76 | 75.6 | 78.0 | -2.4 | 10.5 | 0.83 |
| $WS_{10}$ (m/s) | GZ | 3.1 | 2.4 | 0.7 | 1.9 | 0.75 | 2.6 | 1.8 | 0.8 | 1.8 | 0.68 | 2.8 | 2.4 | 0.4 | 1.3 | 0.76 | 2.4 | 1.8 | 0.6 | 1.4 | 0.74 |
| | HK | 4.3 | 3.3 | 1.0 | 2.3 | 0.74 | 3.6 | 2.7 | 0.9 | 1.7 | 0.68 | 4.2 | 3.3 | 0.9 | 1.8 | 0.76 | 3.2 | 2.7 | 0.5 | 1.4 | 0.74 |
| | NN | 2.5 | 1.3 | 1.2 | 2.3 | 0.73 | 2.3 | 1.5 | 0.8 | 2.1 | 0.68 | 2.0 | 1.3 | 0.7 | 1.5 | 0.75 | 1.9 | 1.5 | 0.4 | 1.2 | 0.74 |
| | SZ | 3.3 | 2.2 | 1.1 | 2.6 | 0.73 | 2.8 | 1.8 | 1.0 | 1.8 | 0.68 | 2.9 | 2.2 | 0.7 | 1.2 | 0.75 | 2.5 | 1.8 | 0.7 | 1.7 | 0.73 |
| $O_3$ (ppb) | GZ | 93.7 | 110.5 | -16.8 | 66.6 | 0.55 | 42.2 | 57.0 | -14.8 | 62.5 | 0.51 | 101.3 | 110.5 | -9.2 | 68.3 | 0.68 | 45.3 | 57.0 | -11.7 | 52.5 | 0.64 |
| | HK | 63.7 | 75.5 | -11.8 | 48.7 | 0.58 | 15.4 | 25.3 | -9.9 | 25.8 | 0.51 | 65.4 | 75.5 | -10.1 | 48.2 | 0.71 | 15.3 | 25.3 | -10.0 | 21.7 | 0.63 |
| | NN | 138.4 | 157.8 | -19.4 | 85.4 | 0.54 | 33.8 | 48.9 | -15.1 | 55.4 | 0.51 | 141.7 | 157.8 | -16.1 | 79.5 | 0.62 | 35.4 | 48.9 | -13.5 | 48.6 | 0.60 |
| | SZ | 64.7 | 80.0 | -15.3 | 54.2 | 0.52 | 28.7 | 43.9 | -15.5 | 50.1 | 0.52 | 67.3 | 80.0 | -12.7 | 56.5 | 0.60 | 31.6 | 43.9 | -12.3 | 41.0 | 0.61 |
| $PM_{10}$ (µg/m³) | GZ | 21.1 | 19.6 | 1.5 | 13.0 | 0.53 | 31.4 | 28.9 | 2.5 | 29.0 | 0.53 | 20.3 | 19.6 | 0.7 | 12.2 | 0.61 | 31.0 | 28.9 | 2.1 | 25.3 | 0.63 |
| | HK | 32.2 | 30.9 | 1.3 | 14.5 | 0.53 | 14.7 | 11.9 | 2.8 | 15.3 | 0.53 | 31.9 | 30.9 | 1.0 | 14.1 | 0.61 | 14.2 | 11.9 | 2.3 | 13.9 | 0.63 |
| | NN | 25.6 | 24.7 | 0.9 | 16.7 | 0.54 | 19.8 | 17.3 | 2.5 | 12.7 | 0.54 | 25.3 | 24.7 | 0.6 | 15.7 | 0.62 | 19.1 | 17.3 | 1.8 | 9.0 | 0.65 |
| | SZ | 27.7 | 28.4 | -0.7 | 14.3 | 0.54 | 24.5 | 20.6 | 3.9 | 17.8 | 0.55 | 28.0 | 28.4 | -0.4 | 13.4 | 0.62 | 23.7 | 20.6 | 3.1 | 14.3 | 0.66 |

[a] Vars indicates the variables, including temperature at 2m ($T_2$), relative humidity at 2m ($RH_2$), wind speed at 10m ($WS_{10}$), ozone ($O_3$) and $PM_{10}$; [b] Site indicates the city where the observation sites locate, including Guangzhou (GZ), Haikou (HK), Nanning (NN) and Shenzhen (SZ); [c] Mean indicates the monthly average value; [d] SIM indicates the simulation results from WRF/Chem; [e] OBS indicates the observation data; [f] COR indicates the correlation coefficients, with statistically significant at 95% confident level.





Table 4 also illustrates the performance of WRF/Chem simulations for the main air pollutants

($O_3$ and $PM_{10}$). Obviously, both Fix_AH and Grd_AH can capture the magnitude and temporal

variation of main air pollutants in these typical cities over South China, and the simulation with

gridded AH fluxes (Grd_AH) can provide better predictions. For Grd_AH, the correlation

coefficients (COR) for $PM_{10}$ in all cities are around 0.62 in January and around 0.65 in July

(statistically significant at the 95 % confident level). The MB values for $PM_{10}$ are only -0.4 - 1.0

$\mu g/m^3$ in January and 1.8 -3.1 $\mu g/m^3$ in July. With respect to $O_3$, the values of MB are -9.2 - -16.1

339       ppb in January and -10.0 - -13.5 ppb in July. These underestimates should be related with the

increasing of $WS_{10}$ and the rising of PBL caused by positive biases in $T_2$. The uncertainties in

emissions of ozone precursors ($NO_x$ and VOCs) may cause these biases as well (Liao et al., 2015;

Xie et al., 2016). However, the values of COR for $O_3$ are 0.60 - 0.71 in January and 0.60 - 0.64 in

July (statistically significant at the 95 % confident level), proving that these modeling results are

reasonable and acceptable.

Fig. 3 presents the monthly-averaged differences of $O_3$ and $PM_{10}$ between Fix_AH and

Grd_AH (Fix_AH minus Grd_AH) at the surface layer over the modeling domain 2 (D02).

Obviously, there are some differences between the two simulations that use different AH

parameterizations. These differences are more obvious in and around big cities because the AH are

related with the human activities. Moreover, the differences in January are higher than those in

July, implying that the adding of AH can arouse more atmospheric disturbances in winter. From

this point of view, Grd_AH can better describe the spatial and temporal heterogeneity of the

impacts of AH on regional air quality.

Above all, the WRF/Chem simulation accounting for the temporal and spatial distribution of

AH (Grd_AH) has a relatively good capability in simulating urban climate and air quality over

South China. So, the differences between the modeling results from Non_AH and Grd_AH can be

used to quantify the impacts of anthropogenic heat on meteorology and air pollution.

**3.3 Impacts of AH on meteorological conditions**

Fig. 4a-d, Fig. 5a-d, Fig. 6a-b and Fig. 6g-h show the impacts of AH on surface meteorology,

which are defined as the monthly-averaged differences of these meteorological factors between

Grd_AH and Non_AH (Grd_AH minus Non_AH) at the surface layer over the modeling domain 2.

Fig. 4e-f and Fig. 6c-f show the relevant vertical changes of the meteorological factors along the

cross-section from (19.1°N, 108.9°E) to (24.8°N, 114.7°E) which is shown as the solid line AB in

Fig. 1b.



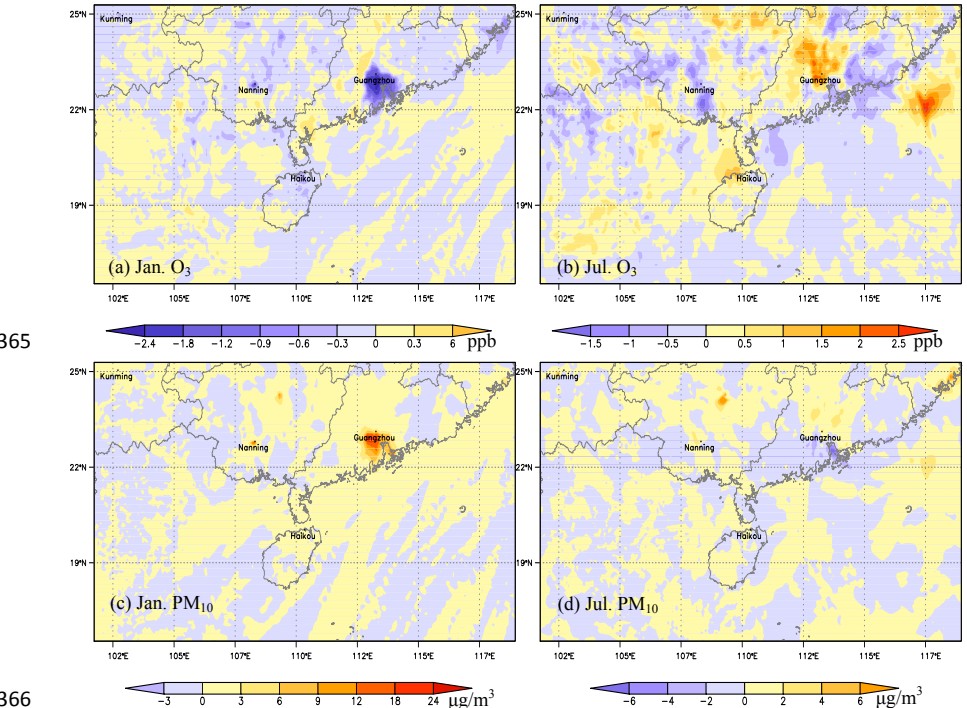

**Fig. 3. The spatial distributions of monthly-averaged differences for surface O₃ and PM₁₀ between Fix_AH and Grd_AH (Fix_AH minus Grd_AH). (a) and (c) show changes in January. (b) and (d) illustrate variations in July.**

### 3.3.1 Changes of surface energy and air temperature

On account that AH and its diurnal variation are added to the sensible heat item in WRF/Chem, the adding of gridded AH fluxes should increase the modeling results of sensible heat fluxes (SHF) over South China. As shown in Fig. 4a and b, the spatial patterns of SHF changes in both January and July are similar to the spatial distribution of AH fluxes presented in Fig. 2f. The significant increments (> 10 W/m$^2$) of SHF over South China usually occur in and around mega-cities. Especially in the PRD city cluster, adding AH can cause SHF to increase by over 50 W/m$^2$ in both January and July.

For the 2-m air temperature (T$_2$) over South China, the AH fluxes can increase their values by adding more surface heat into the atmosphere. As presented in Fig. 4c and d, the patterns of the monthly-averaged T$_2$ changes are similar to those of SHF (Fig. 4a and b). In the urban areas, the adding of AH can lead to the significant increase of T$_2$, which may enhance the Urban Heat Islands. The maximum T$_2$ changes are usually found in the city centers of the PRD region, with the typical increments over 1.1 ℃ in January and over 0.5℃ in July. These findings are comparable to the values estimated for other cities (Fan and Sailor, 2005; Ferguson and Woodbury, 2007; Chen et al., 2009; Zhu et al., 2010; Menberg et al., 2013; Wu and Yang, 2013; Bohnenstengel et al., 2014; Yu




et al., 2014; Xie et al., 2016), and can be confirmed by the similar researches in South China
(Meng et al., 2011; Feng et al., 2012; 2014).






**Fig. 4. The monthly-averaged differences between Grd_AH and Non_AH (Grd_AH minus Non_AH) for (a),**
**(b) the spatial distribution of sensible heat flux (SHF); (c), (d) the spatial distribution of 2-m air temperature**
**($T_2$); (e), (f) the vertical distribution of air temperature (T) from the surface to the 800hPa layer along the**
**line AB shown in Fig. 1b. Grd_AH and Non_AH represent the simulations with and without AH fluxes. (a),**
**(c), and (e) show changes in January, while (b), (d), and (f) illustrate variations in July. In (e) and (f), HK**
**and GZ are the abbreviations for Haikou and Guangzhou, respectively.**

Fig. 4e and f present the vertical changes of air temperature from the surface to the 800hPa





layer along the line AB (shown in Fig. 1b), and illustrate that the increases of air temperature
causing by adding AH are mainly confined near the surface around the cities (Guangzhou and
Haikou). These changes of air temperature in Guangzhou are more obvious than those in Haikou,
because the AH emissions are much higher in Guangzhou. Furthermore, $T_2$ changes in winter (Fig.
4e) are more obvious than those in summer (Fig. 4f), with the monthly mean increment of T over
0.7℃ for January while only around 0.4℃ for July in Guangzhou. This phenomenon should be
related with the fact that the background heat fluxes are much lower in winter so that the relative
increase of T is more obvious.
**3.3.2 Changes of boundary layer and wind field**
The warming up of surface air temperature can enhance the vertical air movement in
boundary layer (PBL), and thereby can increase the height of boundary layer (PBLH) as well. As
shown in Fig. 5a and b, the boundary layer height becomes higher when the AH fluxes are taken
into account. The big increments (more than 50m) usually occur in the urban areas of the PRD
region. Because relative higher temperature increment in January can induce higher PBL in this
cold season, the maximum changing values of PBLH can be 120m for January but only 90m for
July.

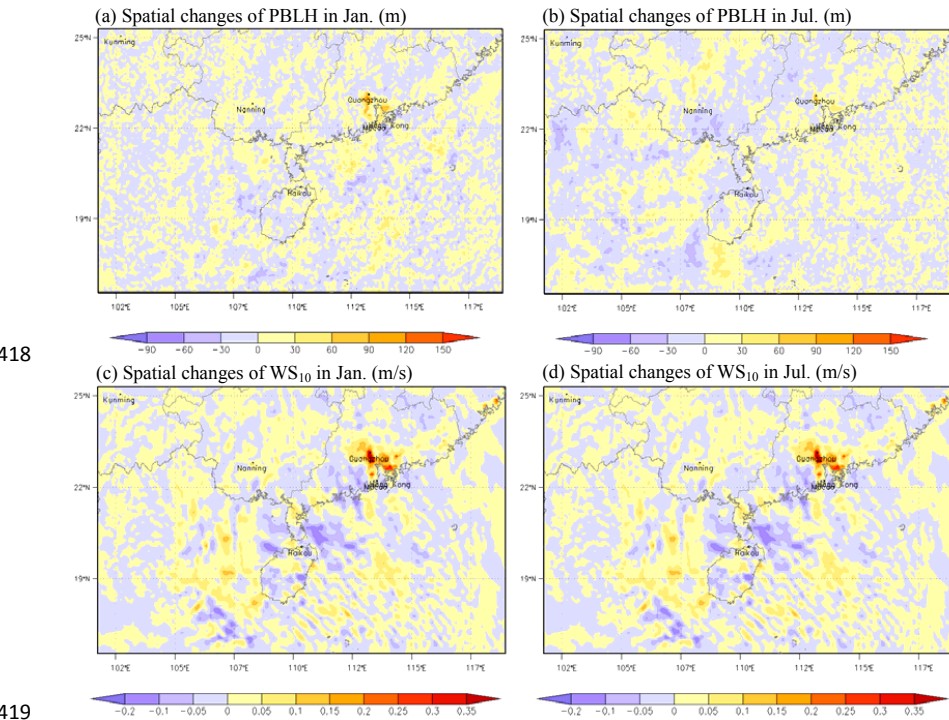



**Fig. 5. The monthly-averaged differences of the height of planetary boundary layer (PBLH) and 10-m wind**
**speed ($WS_{10}$) between Grd_AH and Non_AH (Grd_AH minus Non_AH). Grd_AH and Non_AH represent**
**the simulations with and without AH fluxes. (a) and (c) show changes in January, while (b) and (d) illustrate**





**variations in July.**

Fig. 5c and d show the changes in the 10-m wind speed over South China. Obviously, adding
AH can enhance the surface wind in the urban areas. The maximum increase is located in the PRD
region, with the values over 0.35 m/s in January and 0.3 m/s in July. In other cities like Chaozhou,
Nanning and Haikou, the increments are merely about 0.1 m/s. The warming of air temperature
near surface as well as the rising of PBLH induced by adding AH in cities can generate an
enhanced urban-breeze circulation. In previous studies, the increases in surface wind speed were
considered to be related with this strengthened urban-breeze circulation (Chen et al., 2009; Ryu et
al., 2013; Yu et al., 2014; Xie et al., 2016). Our results show that the vertical wind velocities
above the Guangzhou and Haikou is enhanced in both January and July (Fig. 6c and d), and the
simulated convergence at the surface near these cities increases by 0.04-0.13 /s in January and
0.05-0.18 /s in July (not shown). Consequently, we deduce that the enhanced vertical air
movement causes the surface stronger convergence and thereby induces higher surface wind
speed.
**3.3.3 Changes of moisture and rainfall**
Fig. 6a and b presents the monthly-averaged differences of 2-m relative humidity ($RH_2$)
between Grd_AH and Non_AH. Obviously, the air near the surface of cities becomes dryer. The
negative centers occur in the PRD region, the Chao-Shan area, Haikou and Nanning, which are
also the AH emission centers occurring in Fig. 2f. In and around these cities, the reductions of
surface $RH_2$ are -3 to -4% in January and -1% to -2% in July.
It was reported that the enhanced vertical air movement can transport more moisture from the
surface to the upper layer, and thereby can modify the spatial and vertical distributions of moisture
(Xie et al., 2016). This effect mechanism can be clearly illustrated by Fig. 6c-f in this study. As
shown in Fig. 6c and d, the vertical wind velocities above Guangzhou and Haikou increase by the
values of 0.2 – 0.5 cm/s in January and 0.5 - 1.0 cm/s in July, whereas w decreases in the rural
areas with the reductions about -0.3m/s in January and over -0.5 cm/s in July. This pattern means
that there are a strengthened upward air flow in cities and a strengthened downward air flow in the
surrounding areas, implying that the adding of AH fluxes makes the atmosphere more unstable and
tends to form deep convections in troposphere. So, as shown in Fig. 6e and f, more moisture can
be transported from the surface to the upper layers. In Guangzhou, for example, the water vapor
mixing ratios at the ground level decrease by -0.3g/kg in January and -0.5 g/kg in July, while those
at the upper PBL increase by 0.1 g/kg in January and 0.3 g/kg in July. The impact of AH on water
vapor is stronger in July. This seasonal difference can be ascribed to the facts that the atmosphere
is more stagnant and dryer in winter and more convective and wetter in summer. Furthermore, the
changes in Haikou are generally smaller than those in Guangzhou, which can be explained by the
fact that the AH emissions are much lower in Haikou.





**Fig. 6. The monthly-averaged differences between Grd_AH and Non_AH (Grd_AH minus Non_AH) for (a),**

**(b) the spatial distribution of 2-m relative humidity (RH₂); (c), (d) the vertical distribution of vertical wind**




velocity (w); (e), (f) the vertical distribution of water vapor mixing ratio (VAPOR); (g), (h) the spatial
distribution of precipitation (RAIN). The vertical cross section is from the surface to the 800hPa layer along
the line AB shown in Fig. 1b. Grd_AH and Non_AH represent the simulations with and without AH fluxes.
(a), (c), (e), and (g) show changes in January, while (b), (d), (f), and (g) illustrate variations in July. In (c), (d),
(e), and (f), HK and GZ are the abbreviations for Haikou and Guangzhou, respectively.

More moisture transported from surface into the mid-troposphere can increase the

precipitation in these urban areas as well. Fig. 6g and h illustrate the enhanced rainfall over South
China both in January and July. Because of the negligible accumulative precipitation in winter,
there are no significant differences between the Grd_AH and Non_AH simulations for rainfall in
January. But in July, the increment of rainfall can be more than 50mm in and around big cities.
Moreover, according to the dominant southeast wind in summer, the moisture can be transported
to the downwind areas of the PRD city cluster, which causes the increases of rainfall in the
northwest part of Guangdong province with the maximum value over 80 mm.
**3.3.4 Diurnal pattern of the changes**

In order to better understand the different impacts of AH in the daytime and at night, the

monthly-averaged diurnal variations of $T_2$ and PBLH in January and July over the urban areas in
Guangzhou are calculated based on the results from Grd_AH and Non_AH. As shown in Fig. 7a
and b, adding AH fluxes can lead to an obvious increase of 2-m air temperature in both months,
with the daily mean increase of 1.5℃ for January and 0.6 ℃ for July. The increment of $T_2$ at night
in January (1.69℃) is larger than that in the daytime (1.31℃), whereas the changes during the
whole day in July are all around 0.6℃, which suggests that AH can weaken the diurnal $T_2$
variation in winter. With respect to PBLH, the AH fluxes can also result in a higher boundary layer.
In July (Fig. 7d), the increment of PBLH nearly keeps a constant value of 54m (4.7%) from
morning till night. However, in January (Fig. 7c), the nighttime increase of PBLH is much higher
than that in the daytime. This phenomenon may be related with the facts that the absolute PBLH
values are lower and the air temperatures increase more in the winter nights.



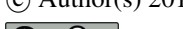

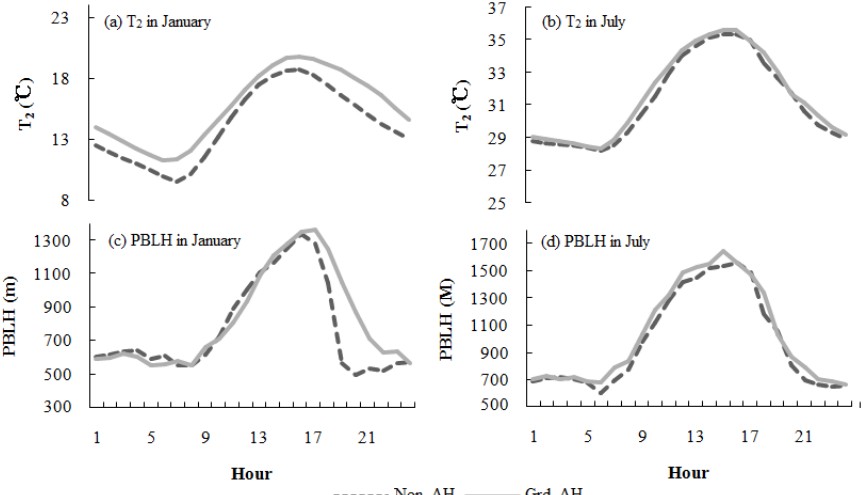

**Fig. 7. The monthly-averaged diurnal variations for 2-m air temperature ($T_2$) and the height of planetary boundary layer (PBLH) over the urban areas in Guangzhou. Grd_AH and Non_AH represent the simulations with and without AH fluxes, respectively. (a) and (c) show diurnal curves in January, while (b) and (d) illustrate those in July.**

**3.4 Impacts of AH on main air pollutants**

**3.4.1 Changes of the spatial and vertical distribution of $PM_{10}$**

Since adding AH changes the meteorology conditions, it can affect the transportation and dispersion of air pollutants as well. Fig. 8a and b show the effects of AH on the spatial distribution of $PM_{10}$ at the surface layer over South China in January and July. They illustrate that the concentrations of $PM_{10}$ decrease in both season near the big cities, including the PRD city cluster, the Chao-shan area, and Nanning etc. The maximum reductions occur in the PRD region, with the monthly mean value over -10μg/m$^3$ for January and about -5μg/m$^3$ for July. Compared with the distribution of AH emissions as well as their effects on meteorological conditions, the main causes resulting in the reduction of surface $PM_{10}$ should be attributed to the increase of PBLH, vertical upward air flow and surface wind speed, which can all facilitate $PM_{10}$ transport and dispersion within the urban boundary layer. For another, as shown in Fig. 6h, the rainfall around the PRD cities can increase by 20-40% in July when the AH fluxes are taken into account, so the strengthened wet scavenging in summer may contribute to the decreases of the surface concentrations of $PM_{10}$ as well. The surface reductions of $PM_{10}$ induced by adding AH in the PRD region are smaller than those reported by Xie et al. (2016) in the Yangtze River Delta (YRD) region, which may attributed to the facts that the particle pollution is more severe and the AH emissions as well as their effects on meteorology are more obvious in the YRD region.



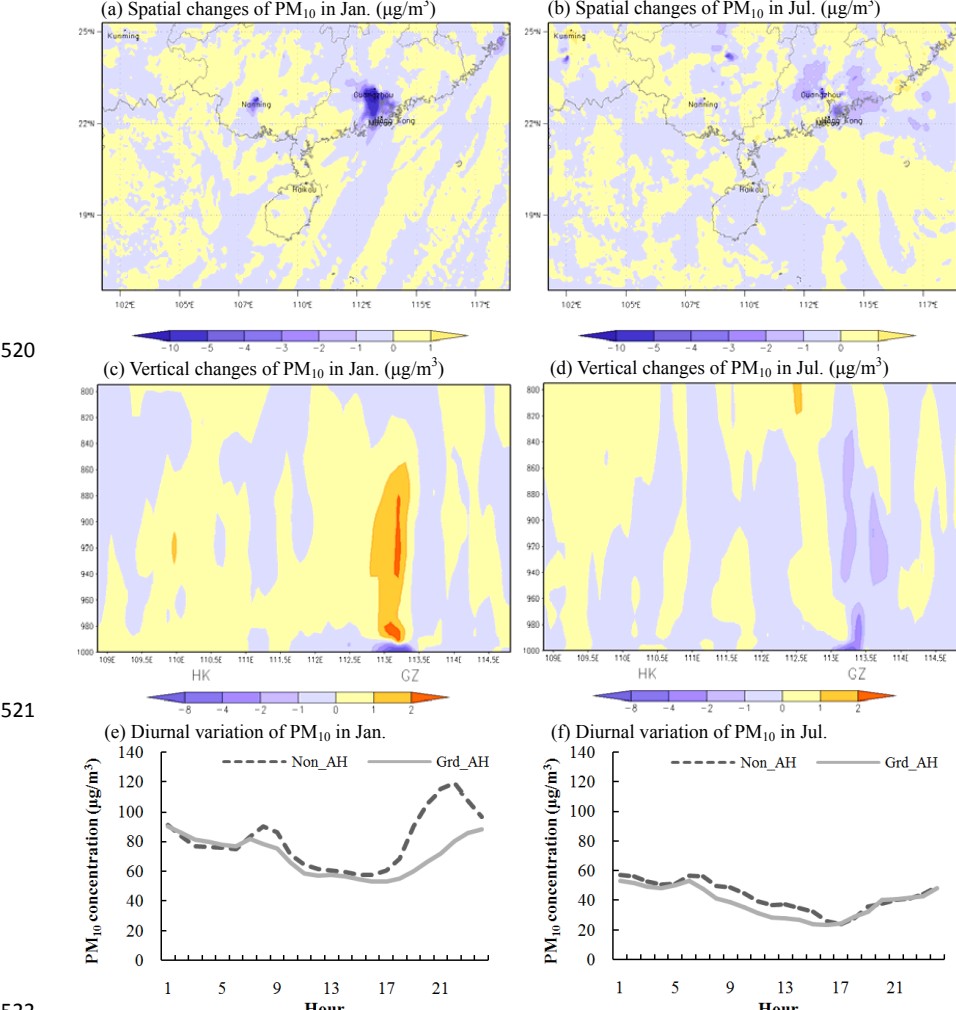

**Fig. 8 Impacts of AH fluxes on the concentrations of PM$_{10}$: (a), (b) the spatial distribution of**
**monthly-averaged differences for PM$_{10}$ between Grd_AH and Non_AH (Grd_AH minus Non_AH) at the**
**surface layer; (c), (d) the vertical distribution of monthly-averaged differences for PM$_{10}$ between Grd_AH**
**and Non_AH (Grd_AH minus Non_AH) from the surface to the 800hPa layer along the line AB shown in**
**Fig. 1b; (e), (f) the monthly-averaged diurnal variations for PM$_{10}$ concentrations over the urban areas in**
**Guangzhou. Grd_AH and Non_AH represent the simulations with and without AH fluxes. (a), (c), and (e)**
**show changes in January, while (b), (d), and (f) illustrate variations in July. In (c) and (d), HK and GZ are**
**the abbreviations for Haikou and Guangzhou, respectively.**
Fig. 8c and d present the vertical plots for the changes of PM$_{10}$ impacted by adding AH
(Grd_AH minus Non_AH) on the cross-sectional line AB shown in Fig. 1b. With respect to the
megacity Guangzhou, the AH fluxes can decrease the concentrations of PM$_{10}$ near surface and
increase those at the upper layers. This vertical change pattern of PM$_{10}$ is quite similar to that of
water vapor (Fig. 6e and f), indicating that it is a reflection of the changes in vertical transport





pattern due to AH (Yu et al., 2014; Xie et al., 2016). As shown in Fig. 8c for January, the decreases
of $PM_{10}$ manly confined at the surface, with the typical reductions over -8μg/m$^3$. Meanwhile, there
are obvious increases of $PM_{10}$ concentrations at the upper levels, with the increments over 2μg/m$^3$
from the 980hPa layer to the 850hPa layer (approximately from 500m to 1500m). But for July (Fig.
8d), from the surface to the 850hPa layer over Guangzhou, the $PM_{10}$ concentrations are all reduced
over -1μg/m$^3$, with the maximum values over -4μg/m$^3$ on the ground. The increasing zones only
occur at the upper layers above 1.5km, with the increments over 1μg/m$^3$. This significant seasonal
difference for the vertical distribution of $PM_{10}$ changes over Guangzhou should be related with the
fact that the atmosphere is more unstable and convective in summer than in winter, which can be
further proven by the phenomenon that the enhanced upward air movement in July is stronger than
that in January (shown in Fig. 6e and f). It should be noted that the vertical changes of $PM_{10}$ in
Haikou are indistinctive, implying that the surface air pollutants cannot be remarkably affected by
adding AH if the heat emission fluxes are less than 10 w/m$^2$. Furthermore, the low particle
pollution level may be another cause for the negligible vertical changes of $PM_{10}$ in Haikou.
Fig. 8e and f show the monthly-averaged diurnal variations of surface $PM_{10}$ from the
Grd_AH and Non_AH simulations over the urban areas in Guangzhou. Obviously, the adding of
AH fluxes can lead to the decrease of surface $PM_{10}$ concentrations, with the daily mean value of
-10.4μg/m$^3$ for January and -4.3μg/m$^3$ for July. There are significant differences between the
impacts of AH in the daytime and those at night. In July (Fig. 8f), the decreases mainly occur from
6:00 to 17:00. In January (Fig. 8e), the decreases are -8.8μg/m$^3$ from 8:00 to 18:00 and -11.9μg/m$^3$
from 19:00 to 7:00, with the maximum reduction of -36.9μg/m$^3$ at 21:00. This pattern has a
reverse correlation with the changes of PBLH shown in Fig. 7c and d, which also manifests the
important role of vertical air movement in the changes of $PM_{10}$.
**3.4.2 Changes of the spatial and vertical distribution of $O_3$**
Fig. 9a and b present the effects of AH on the spatial distribution of $O_3$ at the surface layer
over South China. The results show that the increases of surface $O_3$ level can be seen in megacities
for both January and July. In January (Fig. 9a), the maximum $O_3$ differences occur in the big cities
of the PRD region, with the monthly mean increment over 2.5ppb. In July (Fig. 9b), the increasing
areas become larger, but with the high values close to 1 ppb in and around the cities. This
changing pattern is similar to the findings reported in Seoul (Ryu et al., 2013), Beijing (Yu et al.,
2014) and the cities in the YRD region (Xie et al., 2016).
Fig. 9c and d show the effects of AH on the vertical distribution of $O_3$ from the surface to the
800hPa layer along the line AB (illustrated in Fig. 1b). For the urban areas of Haikou, the vertical
changes of $O_3$ are all within ±0.2 ppb, which means that low AH emissions in this city (<10w/m$^2$)
cannot remarkably affected the physical and chemical formation of $O_3$. However, over the urban
areas of big city Guangzhou, the vertical distribution of $O_3$ concentrations can be noticeably
changed. In January (Fig. 9c), $O_3$ increases at the surface while decreases at the upper levels. The
increases of $O_3$ concentrations are limited within 300m above the surface (<995hPa) over the





urban areas, with the high values over 2.5 ppb. The maximum decreases of $O_3$ concentrations
occur from the 990hPa layer to the 860hPa layer (approximately from 400m to 1500m), and the
typical reductions are about 0.3 ppb. This change pattern in winter for Guangzhou is similar to the
findings reported in Shanghai and Hangzhou (Xie et al., 2016). But for July, the vertical change
pattern of $O_3$ above Guangzhou is totally different. As illustrated in Fig. 9d, $O_3$ concentrations
decrease at the lower layers while increase at the upper levels. The decreases occur from the
surface to the 850hPa layer (about 1.5 km) with the reduction values of -1 to -1.5ppb, and the
increases appear at the upper layers as well as the surrounding air columns around Guangzhou
with the increment about 0.9-1.2 ppb.

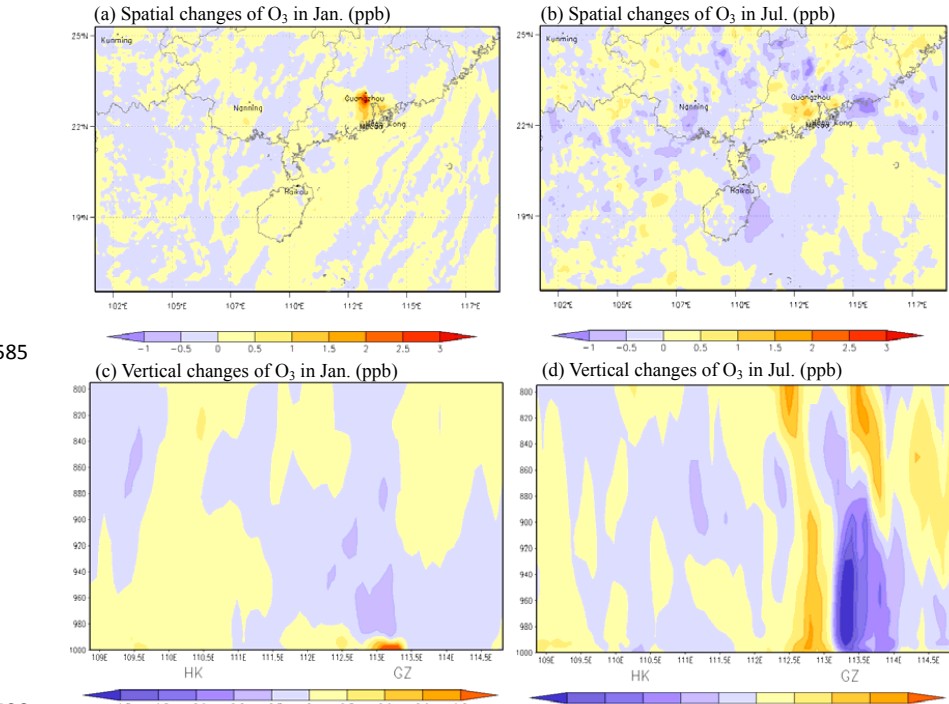


**Fig. 9. Impacts of AH fluxes on the concentrations of $O_3$: (a), (b) the spatial distribution of**
**monthly-averaged differences for $O_3$ between Grd_AH and Non_AH (Grd_AH minus Non_AH) at the**
**surface layer; (c), (d) the vertical distribution of monthly-averaged differences for $O_3$ between Grd_AH and**
**Non_AH (Grd_AH minus Non_AH) from the surface to the 800hPa layer along the line AB shown in Fig. 1b.**
**Grd_AH and Non_AH represent the simulations with and without AH fluxes. (a) and (c) show changes in**
**January, while (b) and (d) illustrate variations in July. In (c) and (d), HK and GZ are the abbreviations for**
**Haikou and Guangzhou, respectively.**

The mechanism how the AH fluxes influence the spatial and vertical distribution of $O_3$ is

more complicated than that for $PM_{10}$. Only taking the physical effects that just impact $O_3$ transport
and dispersion into account, we can merely deduce that $O_3$ is seemingly reduced at the surface and



may increase at the upper layers, because the increase of surface wind speed can facilitate $O_3$
advection transport and the rising up of PBLH can lead to $O_3$ dilution. However, $O_3$ is a secondary
air pollutant produced by a series of complex chemical reactions that are also deeply affected by
the ambient meteorological conditions. So, the chemical effects can play an important role in $O_3$
changes as well. For example, the increases of air temperature induced by adding AH can
accelerate $O_3$ production rate. So it can directly increase the $O_3$ concentrations near the surface
(referred to as the direct chemical effect hereafter). Moreover, because of the $O_3$ sensitivity in the
daytime and the $NO_x$ titration at night, $O_3$ formation is inextricably linked with $NO_x$ (referred to as
indirect chemical effect hereafter). As shown in Fig. 10, due mainly to the increases of PBLH and
upward air flow caused by adding AH, $NO_x$ can decrease at ground level and increase at upper
layers in both January and July. Then when the process of $NO_x$ titration predominate the $O_3$
chemistry at night, less $NO_x$ consumes less $O_3$ and leaves more $O_3$ at the surface while more $NO_x$
consumes more $O_3$ and reduce $O_3$ at the upper layers. For the daytime, because $O_3$ formation is
sensitive to VOC over the cities in South China (Xie et al., 2014), the decrease in surface $NO_x$ can
lead to a slight increase in $O_3$ while the increase of $NO_x$ at upper layers can result in the $O_3$
decrease. In January over Guangzhou, these direct and indirect chemical effects should play a
more important role in $O_3$ changes than the physical effects, and thereby $O_3$ increases at ground
level and decreases at upper layers. But in July, the physical effects should be the governing factor
and cause the different pattern of $O_3$ changes in Guangzhou.

In the previous study on the $O_3$ variations induced by adding AH, it was found that the

vertical changing patterns of $O_3$ over the YRD region in both January and July are always the
same as the pattern shown in the winter of Guangzhou (Xie et al., 2016). Comparing the vertical
changes of w for July in Guangzhou and those in Shanghai or Hangzhou, we can tell that the AH
fluxes can induce stronger upward air movement in the cities of South China, which may be
related with their special topographic and climatic features, and thereby more $O_3$ below the
850hPa layer is transported to the upper layers or to the surrounding areas of Guangzhou. On the
other hand, the rise of air temperature is smaller in Guangzhou than those in the YRD cities, so
there is no enough produced $O_3$ to compensate the loss of $O_3$ on the ground. Consequently,
impacted by adding AH, $O_3$ decreases at the surface while increases at the upper layers in the
summer of Guangzhou.

**4. Conclusions**

Anthropogenic heat (AH) fluxes related with the human activities can change the urban

circulation and thereby affect the air pollution in and around cities. In this paper, we carry out
systematic analyses to study the changes of meteorological conditions induced by AH and their
effects on the concentrations of $PM_{10}$, $NO_x$ and $O_3$ in South China. Firstly, the temporal and spatial
distribution of AH emissions is estimated by a top-down energy inventory method. Secondly, the
AH parameterization in WRF/Chem is modified to adopt the gridded AH data with the temporal



variation. Finally, the WRF/Chem simulations are performed, and the differences between the
cases with and without adding AH are analyzed to quantify the impacts of AH.
The results show that high AH fluxes generally occur in and around the cities. In 2014, the
regional mean values of AH over Guangdong, Guangxi and Hainan province are 1.68, 0.44 and
0.49 $W/m^2$, while the typical values in the urban areas of the PRD region can reach 58.03w/$m^2$.
The model results of WRF/Chem fit the observations well. Adding the gridded AH emissions can
better describe the heterogeneous impacts of AH on regional meteorology and air quality. When
AH fluxes are taken into account, the urban heat island and urban-breeze circulations in the big
cities are significantly changed. In the PRD city cluster, 2-m air temperature rises up by 1.1℃ in
January and over 0.5℃ in July, the boundary layer height increases by 120m in January and 90m
in July, and 10-m wind speed is enhanced over 0.35 m/s in January and 0.3 m/s in July. The
enhanced vertical movement can transport more moisture to higher levels, and causes the
accumulative precipitation to increase by 20-40% over the megacities in July. Influenced by the
modifications of meteorological conditions, the spatial and vertical distribution of air pollutants is
modified as well. The concentrations of $PM_{10}$ and $NO_x$ decrease near surface while increase at the
upper levels over the big cities in the PRD region, which are mainly related with the higher PBLH,
stronger upward air flower, and higher surface wind speed. Because the direct chemical effect (the
rising up of air temperature directly accelerates surface $O_3$ formation) and the indirect chemical
effect (the decrease in $NO_x$ at the ground results in the increase of surface $O_3$) play a more
important role than the physical effects in winter, the surface $O_3$ concentrations can increase in
January with maximum changes over 2.5ppb in the megacities. However, in July, the vertical
changes of $O_3$ concentrations induced by adding AH show a different pattern, with reductions at
the lower layers and increments at the upper layers over Guangzhou. This phenomenon should be
attributed to the fact that the physical effects (enhanced upward movement caused by AH) become
the dominant factor in summer.
There is an important question asked many times by scientists about whether anthropogenic
heat emissions contribute to global warming. Although the answers are probably negative, the
systematic analyses of AH over South China in this paper can enhance the understanding of the
magnitude of AH emission from megacities and its impact on regional meteorology and
atmospheric chemistry. Compared with the effects from urban land use (Wang et al., 2007; 2009b;
Feng et al., 2012; Chen et al., 2014b; Li et al., 2014; 2016; Liao et al., 2015; Zhu et al., 2015), the
impacts of AH are relative small. Especially in some cities with less air pollution and AH
emissions, such as Haikou, the effects of AH on air quality may be ignored. But our results also
clearly show that the meteorology and air pollution predictions in and around big cities are highly
sensitive to the anthropogenic heat inputs. Thus, for further understanding of urban atmospheric
environment issues, more studies of the anthropogenic heat release in megacities should be better
considered.

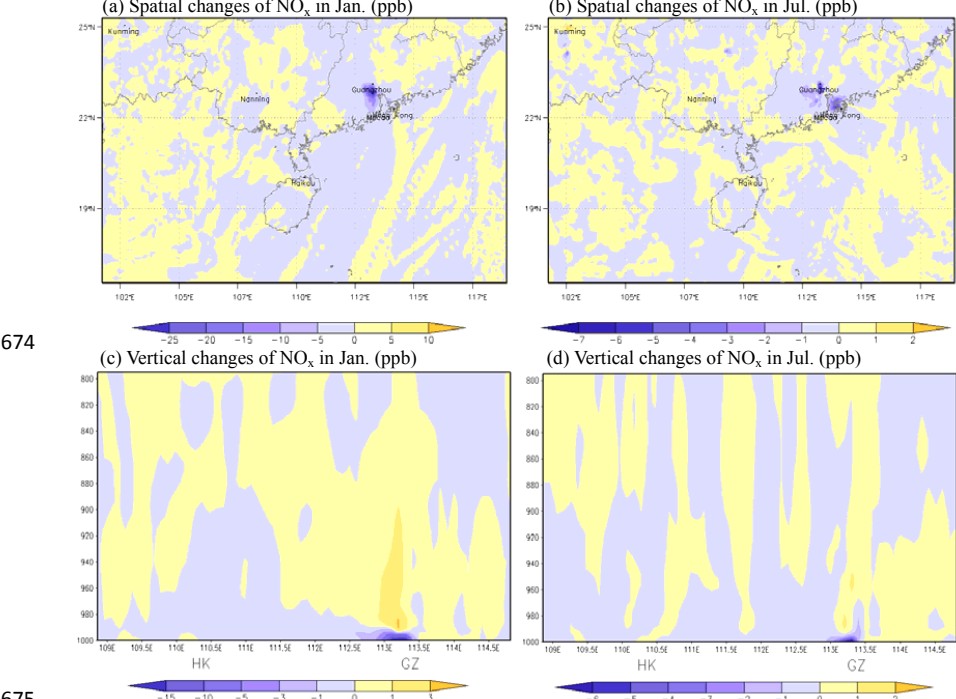



**Fig. 10. Impacts of AH fluxes on the concentrations of NO$_x$: (a), (b) the spatial distribution of monthly-averaged differences for NO$_x$ between Grd_AH and Non_AH (Grd_AH minus Non_AH) at the surface layer; (c), (d) the vertical distribution of monthly-averaged differences for NO$_x$ between Grd_AH and Non_AH (Grd_AH minus Non_AH) from surface to 800 hPa layer along the line AB shown in Fig. 1b. Grd_AH and Non_AH represent the simulations with and without AH fluxes. (a) and (c) show changes in January, while (b) and (d) illustrate variations in July. In (c) and (d), HK and GZ are the abbreviations for Haikou and Guangzhou, respectively.**

### Acknowledgments

This work was supported by the National Natural Science Foundation of China (41475122, 91544230), Key Laboratory of South China Sea Meteorological Disaster Prevention and Mitigation of Hainan Province (SCSF201401), the National Special Fund for Environmental Protection Research in the Public Interest (201409008), the National Science Foundation of Jiangsu Provence (BE2015151), and EU 7th Framework Marie Curie Actions IRSES project REQUA (PIRSES-GA-2013-612671). The authors would like to thank the anonymous reviewers for their constructive and precious comments on this manuscript.

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
