# Peer review of "Changes of regional meteorology induced by anthropogenic heat 1 and their impacts on air quality in South China 2 Min Xie1,2,3\*, Kuanguang Zhu1,5, Tijian Wang1,3,4\*, Wen Feng2, Da Gao1, Mengmeng Li1, Shu Li1, 3 Bingl"

_Atmospheric Chemistry and Physics, 2016_

## Referee Comment (RC1) · Anonymous Referee #1 · 6 Aug 2016

General comments: This is a nice paper on using numerical modeling to address some important issues related to urban effects on the atmosphere. The paper is well written as a whole and easy to follow. The main idea is to analyze the relative importance of anthropogenic heat contribution to atmospheric patterns and its consequent effects on air pollution dispersion. Although the contribution seems to be small compared to other effects, it is clear that it is very important to correctly consider the AH sources during simulations, especially over very large urban areas. I consider the paper suitable for publication, after the verification of some small features, as listed in the specific comments bellow.

Specific Comments and technical corrections:

Abstract: insert the acronym for boundary layer height before using it (see lines 23 and 26).

Page 2, lines 49-50 (and 201). Indicate which reference correspond to Zhang et al 2009. There are two different references with the same year.

Page 5, line 172-173, the authors mention that the model domain has a resolution of 27 km x 27 km. Actually, this refers to the grid spacing, as you mention on line 175. Resolution has another meaning and it is related to the feature that you are able to represent in the model.

The resolution of Fig 1 is to low. It would be better if the authors could improve it.

Page 6, lines 190-191. In the same way, the sentence "the resolution of 30-sec" should be changed by "30 arc seconds grid spacing". Please, consider the same issue on lines 199 and 204.

Page 8, lines 263-264 (and 278) the authors cited the reference Chen et al 2012a. However, there is just on reference by Chen et al in 2012, as you can see in the reference list.

Page 8, line 266. Correct the figure number. The correct is 2f, instead of 4f. In the comparisons of Grd_AH against Fix_AH it would be useful for the readers to have a special view of the Fix_AH contributions. The values as fixed, but they certainly are heterogeneous around the domain. A figure similar to Fig. 2 would be nice.

In page 16, section 3.3.3. The authors mention that the air near to the surface becomes dryer. It should be considered that the regions were RH is lower correspond mostly to the regions were Temperature is higher. Therefore, it not necessarily means that the air is actually drier. I believe that this whole paragraph should be better explained.

Page 19, line 503. The correct statement is "atmospheric conditions" instead of "meteorology conditions".

[Figure]

Page 25, line 652. The authors said that "... strong upward air flower..." Is that right?

References:

Indicate in the list which references corresponds to Chen et al 2014 a and b. The same for Liu et al 2013 a and b, Wang et al 2009 a and b.

Word Bank Group, 2015, cited on page 3, is not correctly cited in the reference list.
* * *

---

## Referee Comment (RC2) · Anonymous Referee #2 · 8 Aug 2016

The article analyses the impacts of anthropogenic heat (AH) emissions on the atmospheric conditions and air quality in South China considering January and July from 2014. The article is based in a spatial and temporal analysis of AH emissions from top-down energy inventory method and WRF/CHEM model simulations.

This paper is very well written, organized, with very clear graphics/figures and with interesting analysis results. Despite of the positive view of the article, there are some deficiencies, but this referee recommends the manuscript to be accepted for publication in the Atmospheric Chemistry and Physics after suggested revisions are made. The suggestions are described below:

Page 3, lines 84-86. The article showed that the main impacts of AH emissions were

observed in Pearl River Delta (PRD) region. However, it is not presented an explanation of that region, such as which cities are located in that area. Please, for readers who do not know that area, provide more detailed information of the area, such as a map illustrating the location. In the same way, the article has a deficiency in the description of the South China region. This referee thinks that is important to describe some information about land use, land cover, topography, as well as the typical climatological and atmospheric conditions such as circulations breezes, among others.

Page 5, Figure 1 - The resolution of Figure 1 is not good. If possible, the authors could improve the figure.

Page 5, lines 176 - 178. In the description of chosen period, that is, January and July from 2014, the paper mentioned that "January and July are used to represent the hot and the cold weather condition, respectively", but the months of January and July represent the cold and hot months for the region analyzed, respectively. Also in this context, why did you choose those periods? Moreover, the use of monthly average could produce erroneous or masked results, since it includes days with different synoptic conditions.

Page 6, lines 182-183. The authors present a vertical cross section analysis through the line AB reaching the Haikou and Guangzhou areas. However, it is not presented the motivation of choosing that line. If line AB was a latitudinal section, approximately 22.5°N or 23°N reaching the Nanning and Guangzhou areas, do you think it would be possible to find a different pattern from the impact of AH emissions? Why?

Page 6, lines 186-193. In the description of the physical parameterizations schemes, it was mentioned about which urban canopy parameters were adopted. Then, it would be interesting to add a descriptive table that contains main urban parameters such as height of buildings and constructions, street and avenues information, albedo of urban areas, among others.

Page 8, line 266. The mentioned figure seems to be wrong (Fig.4f). It would not be

Fig.2f?

Page 12, line 358-360. The article demonstrates the impact of AH emissions on the atmospheric condition through the analysis of some variables such as wind speed at 10m (WS10) and vertical wind velocity (w). Do you think that AH emissions can disturb the horizontal wind regime? How AH emissions can affect the land and the sea breezes circulation? The spatial and temporal patterns of these variables and their correlations would be investigated more properly.

Page 13, line 381-382. The other deficiency is the description that AH emissions can modify the Urban Heat Islands (UHI). It appears to be questionable whether the increase in AH emissions can quantitatively enhance the UHI. The authors could provide concrete evidence of the UHI intensification. One way that authors can analyze could be the temperature difference between the most urbanized region (e.g. Guangzhou) and rural or less urbanized region (e.g. Nanning or Haikou) for simulations with (Grd_AH) and without adding AH (Non_AH). Therefore, perform an analysis of the Urban Heat Island Intensity (or UHII) and examine whether results are in agreement with paper, so, if there is an intensification of the UHI when adding AH.
* * *

---

## Author Comment (AC1) · 28 Oct 2016

General comments: This is a nice paper on using numerical modeling to address some important issues related to urban effects on the atmosphere. The paper is well written as a whole and easy to follow. The main idea is to analyze the relative importance of anthropogenic heat contribution to atmospheric patterns and its consequent effects on air pollution dispersion. Although the contribution seems to be small compared to other effects, it is clear that it is very important to correctly consider the AH sources during simulations, especially over very large urban areas. I consider the paper suitable for publication, after the verification of some small features, as listed in the specific comments bellow.

Thanks for the constructive and the affirmative comments.

Specific Comments and technical corrections: Abstract: insert the acronym for boundary layer height before using it (see lines 23 and 26).
Thanks for the constructive comment. In the new revised manuscript, the acronym 'PBLH' is inserted after the words 'planetary boundary layer height' on line 26.

Page 2, lines 49-50 (and 201). Indicate which reference correspond to Zhang et al 2009. There are two different references with the same year.
Yes, there are two different references with the same year for 'Zhang et al., 2009' in the original manuscript. In the new revised manuscript, they are cited as 'Zhang et al., 2009a' and 'Zhang et al., 2009b', respectively. 'Zhang et al., 2009' on lines 49-50 of the original manuscript is changed to 'Zhang et al., 2009a' in the revised manuscript (lines 48-49). 'Zhang et al., 2009a' is written by Zhang D. L. et al. in 2009, which is listed on lines 887-888 in References of the revised manuscript. 'Zhang et al., 2009' on line 201 of the original manuscript is changed to 'Zhang et al., 2009b' in the revised manuscript (lines 205-206). This paper is written by Zhang Q. et al. in 2009, which is listed on lines 889-891 in References of the revised manuscript.

Page 5, line 172-173, the authors mention that the model domain has a resolution of 27 km x 27 km. Actually, this refers to the grid spacing, as you mention on line 175. Resolution has another meaning and it is related to the feature that you are able to represent in the model.
Thanks for the constructive comment. The words 'with the grid resolution of 27km $\times$ 27km' on lines 172-173 of the original manuscript are replaced by the words 'with the grid spacing of 27km'. Please see line 179 in the new revised manuscript.

The resolution of Fig 1 is to low. It would be better if the authors could improve it.
Thanks for the constructive comment. Fig. 1 is replaced by a new high quality figure, with the improved resolution of 600 dpi. Please see lines 212-217 in the new revised manuscript.
Page 6, lines 190-191. In the same way, the sentence "the resolution of 30-sec" should be changed by "30 arc seconds grid spacing". Please, consider the same issue on lines 199 and 204.

According to the suggestion, the sentence "the resolution of 30-sec" on lines 190-191 of the original manuscript is changed to "30 arc seconds grid spacing", the words "with $0.25° \times 0.25°$ resolution" on line 199 of the original manuscript is revised to "with $0.25°$ grid spacing", and the words "with the spatial resolution of $1° \times 1°$" on line 204 of the original manuscript is replaced by "with the grid spacing of $1°$". Please see lines 194-195, lines 203-204, and line 208 in the new revised manuscript.

Page 8, lines 263-264 (and 278) the authors cited the reference Chen et al 2012a. However, there is just on reference by Chen et al in 2012, as you can see in the reference list.

Sorry for this clerical mistake. The references cited on lines 263-264 of the original manuscript ("Chen et al. (2012; 2014a)") should be "Chen et al. (2012; 2014a)". We change the relevant words, and revise the related label of the papers in the reference list as well. Please see lines 276-277, and lines 751-755 of the new revised manuscript.

Page 8, line 266. Correct the figure number. The correct is 2f, instead of 4f. In the comparisons of Grd-AH against Fix-AH it would be useful for the readers to have a special view of the Fix-AH contributions. The values as fixed, but they certainly are heterogeneous around the domain. A figure similar to Fig. 2 would be nice.

Sorry for this clerical mistake. The figure number "4f" on line 266 of the original manuscript is replaced by "2f". Please see line 279 of the new revised manuscript.

We agree that "the comparisons of Grd-AH against Fix-AH would be useful for the readers to have a special view of the Fix-AH contributions". So, Fig. 1b that is similar to Fig. 2 and some words are added to illustrate and compare the heterogeneity of AH distribution in Grd-AH and Fix-AH. Please see lines 212-217 and lines 288-295.

In page 16, section 3.3.3. The authors mention that the air near to the surface becomes dryer. It should be considered that the regions were RH is lower correspond mostly to

the regions were Temperature is higher. Therefore, it not necessarily means that the air is actually drier. I believe that this whole paragraph should be better explained.

We agree that "the regions where RH is lower mostly correspond to the regions where temperature is higher", and RH2 is not the appropriate factor to explain whether the surface becomes dryer. So, in the new revised manuscript, we use the changes of water vapor mixing ratio at 2m between Grd-AH and Non-AH (Grd-AH minus Non-AH) to explain the effect of AH on surface moisture. We believe that the new figures (Fig. 6a and b) and new paragraph (lines 465-470) in the revised manuscript can better illustrate the phenomenon that the air near the surface becomes dryer.

Page 19, line 503. The correct statement is "atmospheric conditions" instead of "meteorology conditions".

According to the suggestion, the words "meteorology conditions" in "Since adding AH changes the meteorology conditions" on line 503 of the original manuscript are changed to "atmospheric conditions". Please see line 534 of the new revised manuscript.

Page 25, line 652. The authors said that ". . . strong upward air flower. . ." Is that right?

Sorry for this clerical mistake. The words "strong upward air flower" on line 652 of the original manuscript are revised to "strong upward air flow". Please see line 683 of the new revised manuscript.

References: Indicate in the list which references corresponds to Chen et al 2014 a and b. The same for Liu et al 2013 a and b, Wang et al 2009 a and b.

Two different references for 'Chen et al., 2014' are indicated as "Chen et al., 2014a" for the paper written by Chen, B., Dong, L., Shi, G. Y., et al. and "Chen et al., 2014b" for the paper written by Chen, B., Yang, S., Xu, X. D., et al., respectively. Please see lines 751-755 of the new revised manuscript. Meanwhile, the words "Chen et al., 2012a; 2014" on line 278 of the original manuscript are changed to "Chen et al., 2012; 2014a". Please see line 289 of the new revised manuscript.

Two different references for "Liu et al., 2013" are indicated as "Liu et al., 2013a" for the paper written by Liu, M., et al. and "Liu et al., 2013b" for the paper written by Liu, Q., et al., respectively. Please see lines 813-818 of the new revised manuscript. Meanwhile, the words "Liu et al., 2013" on line 164 of the original manuscript are revised to "Liu et al., 2013b" on line 171 of the new revised manuscript. The words "Following the work of Liu et al. (2013)" are replaced by "Following the work of Liu et al. (2013b)" on line 190 of the new revised manuscript.

Two different references for "Wang et al., 2009" are indicated as "Wang et al., 2009a" for the paper written by Wang T., et al. and "Wang et al., 2009b" for the paper written by Wang X. M., et al., respectively. Please see lines 861-866 of the new revised manuscript.

Word Bank Group, 2015, cited on page 3, is not correctly cited in the reference list.

In the reference list of revised manuscript, the cited document is corrected as "World Bank Group: East Asia's changing urban landscape: measuring a decade of spatial growth, World Bank, Washington Dc, 2015.". Please see lines 870-871 of the new revised manuscript.

---

## Author Comment (AC2) · 28 Oct 2016

The article analyses the impacts of anthropogenic heat (AH) emissions on the atmospheric conditions and air quality in South China considering January and July from 2014. The article is based in a spatial and temporal analysis of AH emissions from top-down energy inventory method and WRF/CHEM model simulations. This paper is very well written, organized, with very clear graphics/figures and with interesting analysis results. Despite of the positive view of the article, there are some deficiencies, but this referee recommends the manuscript to be accepted for publication in the Atmospheric Chemistry and Physics after suggested revisions are made. The suggestions are described below:

Thanks for the constructive and the affirmative comments.

Page 3, lines 84-86. The article showed that the main impacts of AH emissions were observed in Pearl River Delta (PRD) region. However, it is not presented an explanation of that region, such as which cities are located in that area. Please, for readers who do not know that area, provide more detailed information of the area, such as a map illustrating the location. In the same way, the article has a deficiency in the description of the South China region. This referee thinks that is important to describe some information about land use, land cover, topography, as well as the typical climatological and atmospheric conditions such as circulations breezes, among others.

Thanks for the constructive comment. To provide more detailed information of South China and PRD, Fig. 1b (with the green square to show the location of PRD) and some words (to briefly describe the information of topography, land use, climate, and atmospheric conditions, etc.) are added in the new revised manuscript. Please see lines 79-93 (brief description) and lines 212 -217 (Fig. 1b) in the revised manuscript.

Page 5, Figure 1 - The resolution of Figure 1 is not good. If possible, the authors could improve the figure.

Thanks for the constructive comment. Fig. 1 is replaced by a new high quality figure, with the improved resolution of 600 dpi. Please see lines 212-217 in the new revised manuscript.

Page 5, lines 176 - 178. In the description of chosen period, that is, January and July from 2014, the paper mentioned that "January and July are used to represent the hot and the cold weather condition, respectively", but the months of January and July represent the cold and hot months for the region analyzed, respectively. Also in this context, why did you choose those periods? Moreover, the use of monthly average

could produce erroneous or masked results, since it includes days with different synoptic conditions.

Sorry for the clerical mistake "January and July are used to represent the hot and the cold weather condition, respectively". These words are changed to "In South China, January is generally representative of the relatively cold and dry season, while July represents the relatively hot and wet weather condition" in the new revised manuscript. Please see lines 183-186.

In the paper for studying the influence of urban expansion on O3 distribution over the PRD region (Wang X. M. et al., 2014), it is reported that "Representations of seasonal results are created using hourly URB results from January and July. The two months are representative of the relatively cold and dry season of the year, and the relatively hot and wet season of the year, respectively. ". So, we choose January and July of 2014 for our simulations. To better clarify our consideration, we rewrite the relevant sentences and cite the paper by Wang et al. (2014) in the new revised manuscript. Please see lines 183-186 and lines 867-869.

In previous studies, Ryu et al. (2013) studied the effects of AH based on an episode, while Yu et al. (2014) investigated this issue by using the monthly average (August) as well. We agree that "the use of monthly average could produce erroneous or masked results, since it includes days with different synoptic conditions.". But the main purpose of this paper is not to discuss the effect of AH on a pollution episode. We want to know the relative longtime effect of AH, its tendency, and the seasonal difference. In this case, it is a common method to use the monthly mean values to discuss the effect (Wang et al., 2014; Yu et al., 2014; Liao et al., 2015; Xie et al., 2016).

Page 6, lines 182-183. The authors present a vertical cross section analysis through the line AB reaching the Haikou and Guangzhou areas. However, it is not presented the motivation of choosing that line. If line AB was a latitudinal section, approximately

22.5âŮęN or 23âŮęN reaching the Nanning and Guangzhou areas, do you think it would be possible to find a different pattern from the impact of AH emissions? Why?

The vertical cross section analysis through the line AB is to discuss the different effects of AH on ambient environment between the big (Guangzhou) and the relatively small (Haikou) city. To better present the motivation of choosing this line, we add these words for explanation on lines 380-381 of the new revised manuscript.

We choose Haikou as the representative of relatively small cities because there are no other cities between Guangzhou and Haikou along line AB.

The AH emission in Haikou is close to that in Nanning. So, we believe that the vertical changing pattern from the impact of AH should be similar if line AB reaches Nanning and Guangzhou. We also do the vertical cross section analysis through the line reaching the Guangzhou and Nanning areas. The results are similar. For example, the figures to illustrate the vertical changes of O3 impacted by adding AH (Grd-AH minus Non-AH) are similar to Fig. 9c and d.

Page 6, lines 186-193. In the description of the physical parameterizations schemes, it was mentioned about which urban canopy parameters were adopted. Then, it would be interesting to add a descriptive table that contains main urban parameters such as height of buildings and constructions, street and avenues information, albedo of urban areas, among others.

Thanks for the constructive comments. We add the descriptive table (Table 2) that contains the modified values of main urban parameters. Please see line 221 of the new revised manuscript. Additionally, we also add some explanation words for the table on lines 190-193.

Page 8, line 266. The mentioned figure seems to be wrong (Fig.4f). It would not be

Fig.2f?

Sorry for this clerical mistake. The figure number "4f" on line 266 of the original manuscript is replaced by "2f". Please see line 279 of the new revised manuscript.

Page 12, line 358-360. The article demonstrates the impact of AH emissions on the atmospheric condition through the analysis of some variables such as wind speed at 10m (WS10) and vertical wind velocity (w). Do you think that AH emissions can disturb the horizontal wind regime? How AH emissions can affect the land and the sea breezes circulation? The spatial and temporal patterns of these variables and their correlations would be investigated more properly.

We agree that AH emissions may affect the land and the sea breeze circulation. We also think that it is a good idea to study the influence of AH on these local breezes. We add "It is worth mentioning that the changes of vertical air movement and surface wind may affect the local land-sea breeze circulation in the coastal cities. For example, AH emission in Haikou enhances the upward air movement above the city (Fig. 6c and d), causes the downward movement above the surrounding waters (Fig. 6c and d), and increases the surface wind from sea to land (stronger convergence). These changes imply that AH might strengthen sea breeze in the daytime and weaken land breeze at night." in Section 3.3.2. We also add Fig. 7e and f to discuss the temporal pattern of the effect of AH on WS10, and find that "For WS10, AH emission causes it to increase 0.07 m/s in January and 0.15m/s in July. Most increases occur in the daytime. The effect of AH on surface wind is negligible at night, which may be related to the fact that the land breeze at night (from land to sea) hinders the surface convergence (from sea to land) caused by AH. ". Please see lines 458-463 and 520-524 in the new revised manuscript.

To perfectly discuss this issue, we should focus on a smaller region and use high-resolution simulations, which we plan to do in the future.

Page 13, line 381-382. The other deficiency is the description that AH emissions can modify the Urban Heat Islands (UHI). It appears to be questionable whether the increase in AH emissions can quantitatively enhance the UHI. The authors could provide concrete evidence of the UHI intensification. One way that authors can analyze could be the temperature difference between the most urbanized region (e.g. Guangzhou) and rural or less urbanized region (e.g. Nanning or Haikou) for simulations with $(Grd_AH) and without adding AH (Non_AH). Therefore, perform an analysis of the Urban Heat Island Intensity (or UHII) and e$

$Thanks for the constructive comments. We perform an analysis of the Urban Heat Island Intensity, and find that AH emission.$